# Detection of Dengue Virus 1 and Mammalian Orthoreovirus 3, with Novel Reassortments, in a South African Family Returning from Thailand, 2017

**DOI:** 10.3390/v16081274

**Published:** 2024-08-09

**Authors:** Petrus Jansen van Vuren, Rhys H. Parry, Janusz T. Pawęska

**Affiliations:** 1Australian Centre for Disease Preparedness, CSIRO Australian Animal Health Laboratory, Private Bag 24, Geelong, VIC 3220, Australia; 2Centre for Emerging Zoonotic and Parasitic Diseases, National Institute for Communicable Diseases, National Health Laboratory Service, Johannesburg 2131, South Africa; paweskajanusz46@gmail.com; 3School of Chemistry and Molecular Biosciences, University of Queensland, Brisbane, QLD 4072, Australia; r.parry@uq.edu.au; 4Centre for Viral Zoonoses, Department of Medical Virology, Faculty of Health Sciences, University of Pretoria, Pretoria 0001, South Africa; 5Faculty of Health Sciences, School of Pathology, University of Witwatersrand, Johannesburg 2050, South Africa

**Keywords:** dengue virus, mammalian orthoreovirus, MRV, imported infections, clinical metagenomics, metagenomics

## Abstract

In July 2017, a family of three members, a 46-year-old male, a 45-year-old female and their 8-year-old daughter, returned to South Africa from Thailand. They presented symptoms consistent with mosquito-borne diseases, including fever, headache, severe body aches and nausea. Mosquito bites in all family members suggested recent exposure to arthropod-borne viruses. Dengue virus 1 (Genus *Orthoflavivirus*) was isolated (isolate no. SA397) from the serum of the 45-year-old female via intracerebral injection in neonatal mice and subsequent passage in VeroE6 cells. Phylogenetic analysis of this strain indicated close genetic identity with cosmopolitan genotype 1 DENV1 strains from Southeast Asia, assigned to major lineage K, minor lineage 1 (DENV1I_K.1), such as GZ8H (99.92%) collected in November 2018 from China, and DV1I-TM19-74 isolate (99.72%) identified in Bangkok, Thailand, in 2019. Serum samples from the 46-year-old male yielded a virus isolate that could not be confirmed as DENV1, prompting unbiased metagenomic sequencing for virus identification and characterization. Illumina sequencing identified multiple segments of a mammalian orthoreovirus (MRV), designated as Human/SA395/SA/2017. Genomic and phylogenetic analyses classified Human/SA395/SA/2017 as MRV-3 and assigned a tentative genotype, MRV-3d, based on the S1 segment. Genomic analyses suggested that Human/SA395/SA/2017 may have originated from reassortments of segments among swine, bat, and human MRVs. The closest identity of the viral attachment protein σ1 (S1) was related to a human isolate identified from Tahiti, French Polynesia, in 1960. This indicates ongoing circulation and co-circulation of Southeast Asian and Polynesian strains, but detailed knowledge is hampered by the limited availability of genomic surveillance. This case represents the rare concurrent detection of two distinct viruses with different transmission routes in the same family with similar clinical presentations. It highlights the complexity of diagnosing diseases with similar sequelae in travelers returning from tropical areas.

## 1. Introduction

The global movement of people significantly contributes to the importation of exotic viral diseases into non-endemic regions. This phenomenon is particularly pronounced in the global spread of arboviruses, viruses transmitted by arthropods such as mosquitoes and ticks. Dengue virus (DENV) is a leading arthropod-borne virus affecting humans worldwide [1]. Dengue fever, caused by DENV, is a significant public health concern, affecting an estimated 390 million people each year and endemic to 120 countries, primarily in Southeast Asia, South Asia, and South America [2]. Dengue fever typically manifests with high fever, severe headache, myalgia, arthralgia, nausea, emesis, and cutaneous rashes. Severe cases escalate to dengue hemorrhagic fever with increased bleeding, thrombocytopenia, and plasma leakage [3,4].

Dengue virus is transmitted to humans primarily via the bite of female mosquitoes of the genus *Aedes* [5]. From 2000 to 2016, 176 laboratory-confirmed dengue cases were identified in returned travelers, underscoring the risk of disease importation through international travel in South Africa [6]. Historically, South Africa has experienced at least three outbreaks of autochthonous dengue transmission, such as those in KwaZulu-Natal in 1897, 1901 and notably in 1926/1927, which resulted in 50,000 cases and 60 fatalities. However, DENV did not become endemic following these outbreaks [6].

Mammalian orthoreovirus (MRV) is a non-enveloped, segmented double-stranded RNA virus of the family *Reoviridae*, genus *Orthoreovirus* [7]. MRVs are generalist and ubiquitous pathogens with a broad mammalian host range and are divided into three major genotypes and serotypes (MRV-1, MRV-2, and MRV-3). A fourth serotype (MRV-4) exists with only one isolate, Ndelle reovirus [8]. MRV typing is primarily distinguished by the S1 segment, which encodes viral attachment protein σ1 and non-structural protein σ1s, believed to govern tissue tropism and host range [9]. Three serotypes have been delineated based on the capacity of type-specific antisera to neutralize virus infectivity and inhibit hemagglutination. The prototype isolates are type 1 Lang (T1L), type 2 Jones (T2J), type 3 Dearing (T3D) and Abney (T3A).

The MRV genome comprises ten linear double-stranded RNA (dsRNA) segments in equimolar ratios within the virion. The segments have terminal non-translated regions that vary in length, and the major open reading frames (ORFs) range from 1059 to 3867 base pairs. The untranslated regions are compatible with other serotypes, allowing for extensive inter-serotype reassortment and the generation of mosaics of viruses with assorted segments from bat, porcine, bovine, and human hosts.

The zoonotic potential of MRVs is notable, as they are commonly spread via fecal-oral or respiratory routes and are frequently found in wastewater surveillance worldwide [10,11,12]. In South Africa, a divergent *Orthoreovirus* was identified from *Eucampsipoda africana* bat flies [13], but neither have MRVs been reported nor have serologic surveillance studies been undertaken. MRVs are usually associated with asymptomatic or mild respiratory [14,15] and enteric infections such as acute or persistent gastroenteritis [16,17,18,19,20]. Severe cases, however, have been documented in children associated with necrotizing encephalopathy [21] and meningitis [22]. MRVs have been identified extensively in pigs and bats in Europe [23,24,25], and generally, MRV causes limited pathology in infected bats [26]. Two independent experimental infection studies on MRV3 in pigs have demonstrated 100% mortality in neonatal pigs, with the development of acute gastroenteritis and severe diarrhea within 72 h of infection [27,28].

Here, we report on the isolation and genetic characterization of DENV1 and MRV3 from a family returning to South Africa from Thailand in July 2017. This case represents a rare instance of concurrent detection of two different viruses with distinct transmission routes, highlighting the complexity associated with diagnosing tropical diseases in returning travelers, and the impact of global travel on pathogen spread

## 2. Materials and Methods

### 2.1. Sample Preparation, Virus Isolation and Sequencing

Clotted blood samples were submitted to the Arbovirus Reference Laboratory, Centre for Emerging, Zoonotic and Parasitic Diseases, National Institute for Communicable Diseases, in Sandringham for arbovirus investigations. The serum was separated via centrifugation. Serology for arbovirus antibodies was performed using a hemagglutination inhibition screening assay [29] to detect antibodies to both endemic and exotic arboviruses, including dengue, chikungunya, West Nile and Sindbis. Virus isolation via the intracerebral inoculation of suckling mice is routinely performed on suspected arbovirus cases at the Arbovirus Reference Laboratory as part of passive arbovirus surveillance for existing, new and emerging viruses.

Virus isolation was attempted via the intracerebral inoculation of suckling mice (NIH strain) with the patient serum. After mice succumbed to virus infection, brain homogenates were clarified and passaged in VeroE6 cells. For the SA397 sample, the intracerebral inoculation of suckling mice for two passages was performed and passaged once on VeroE6 (ATCC No. CRL-1586) green monkey kidney cells. For the SA397 sample, intracerebral inoculation of suckling mice was undertaken for three passages and two passages of VeroE6 cells, as previously described [30].

Viral RNA was extracted from the clarified supernatants and mouse homogenates using Qiagen Viral RNA Mini Kit (Qiagen, Hilden, Germany). Virus cDNA was prepared as described before [31]. Sequencing libraries were prepared using the Nextera DNA library preparation kit recommended by the manufacturer (Illumina, San Diego, CA, USA) and sequenced on the MiSeq Illumina platform. Random hexamer and adapter sequences were removed from the reads using fastp (v0.23.2) [32] under default settings. Clean reads were assembled using SPAdes (v3.15.5, --rnaviral flag), and viral contigs were identified using BLASTx (BLAST+ v2.15.0) as previously described [33]. SPAdes assembled DENV-1 and MRV-3 genome segments were remapped to the reference using BWA-MEM (v0.7.13-r1126) with default settings, and coverage was manually inspected using Integrative Genomics Viewer (v2.3) [34]. Nucleotide sequences determined in this study were screened for vector contamination using NCBI VecScreen (https://www.ncbi.nlm.nih.gov/tools/vecscreen/, UniVec build #10.1, accessed on 17 November 2023) and were deposited in GenBank under the accession numbers MRV: PP953510-PP953519 and DENV1 PQ097697.

### 2.2. Phylogenetic Analysis

For phylogenetic placement of the DENV1 strain, 43 DENV1 genotype I (major lineages A, B, D, H, J, K), 4 genotype II, 3 genotype III, 6 genotype IV, 3 genotype V, and 4 genotype VI were downloaded from Genbank and aligned using MAFFT (v7.475, L-INS-i method). The DENV subtyping tool, as per the new Hill et al., 2024, nomenclature [35], was used to identify the genotypes and lineages with GenomeDetective (https://www.genomedetective.com/app/typingtool/dengue/, accessed on 2 July 2024) [36,37]. For the phylogenetic placement of the S1 segment of the SA strain, 41 strains of MRV3, MRV2 (10 isolates), and MRV1 (8 isolates) were aligned, and metadata such as collection date, host collection and country of origin were taken from Genbank or original publications. As MRV is a studied tool for oncolytic therapy [38], all isolates passed extensively in cells or adapted to cell lines were excluded from the phylogenetic inference. The resultant DENV1 and MRV S1 multiple-sequence alignments were used to construct a consensus maximum-likelihood phylogenetic tree using IQ-TREE2 (v2.1.2), using the GTR+F+G4 nucleotide substitution model for (MORV S1) and TIM2+F+G4 (DENV1) as selected using the Bayesian Information Criterion in the IQ-TREE2 ModelFinder [39]. The resultant consensus tree from combined bootstrap trees was visualized using FigTree v1.4 (software available at http://tree.bio.ed.ac.uk/software/figtree/, accessed on 2 July 2024).

## 3. Results

### 3.1. Case Presentation

In July 2017, a family of three from the Western Cape who had recently returned to South Africa from travel in Thailand between 30 June 2017 and 16 July 2017 presented with symptoms consistent with mosquito-borne diseases (Table 1). The date of onset of symptoms was only recorded for one patient (SA397) as 16 July (date of the family’s arrival back in South Africa), and blood samples were collected from all three patients on 17 July (one day after arrival in South Africa). The family consisted of a 46-year-old male (SA395), an 8-year-old female (SA396), and a 45-year-old female (SA397). They all reported fever along with severe body aches. Patient SA396 also experienced nausea. Each family member confirmed mosquito bites during their travel. The absence of a rash, a common sign in many mosquito-borne illnesses, was noted in all three cases, and no one was hospitalized. Serology towards arboviruses using hemagglutination inhibition assays (HAI) revealed no HAI inhibition against DENV in any samples, which aligns with the tendency for antibodies to not yet be detectable during the acute phase of infection (patient SA397). The only positive in HAI was a low titer of antibodies in SA396 against the Chikungunya virus (CHIKV, *Alphavirus*), but this was not further investigated due to no virus being isolated from the patient sample

### 3.2. Assembly and Phylogenetic Position of Imported DENV-1 Genome

Serum samples from patient SA397 were propagated in suckling mice for two passages and once in VeroE6 cells, and cross-referenced with deep sequencing conducted on RNA isolated from the serum directly. Unbiased deep sequencing and de novo assembly revealed a single contig 10,649 nt in length with a coding-complete 3392 amino acid polyprotein sequence and recovery of 94 and 379 nt of the 5′ and 3′ UTRs. BLASTn of the recovered DENV1 contig against deposited strains that revealed the virus was most closely related in terms of nucleotide identity (99.92%) to an unpublished DENV1 GZ8H/2019002/2018 human isolate (GenbankID: MN869904) collected on the 30th of November 2018 in China, and that it showed 99.72% identity with the DV1I-TM19-74 isolate identified in Bangkok, Thailand, in 2019. Phylogenetic inference of the imported case with other closely related DENV1 sequences revealed that the imported strain clustered with major lineage K, minor lineage 1 (DENV1I_K.1) isolates from Southeast Asia (Figure 1), with most closely related strains from Thailand, China and Cambodia [40]. It is noted that many of the closely related DENV1 sequences have metadata that indicate that they are also cases imported into China from Thailand [41].

### 3.3. Assembly and Phylogenetic Position of MRV Genome

Serum samples from patient SA395 were propagated in suckling mice for three passages and three times in VeroE6 cells and subjected to unbiased deep sequencing for the second and third VeroE6 passages. De novo assembly of the combined sequencing data revealed an almost complete MRV3 virus with coding-complete segments, of which 9 segments recovered (Table 1) and 3197/3929 (81%) of the L2 segment recovered (Figure 2A). We recovered complete sequences, including conserved terminal sequences 5′-GCUA and UCAUC-3′ for M2/3 and S3/4 segments. BLASTn sequence analysis of the assembled segments with their closest relative cross-referenced with the MRV serotype that the closely related segment has been identified from suggests that Human/SA395/SA/2017 MRV-3 may have arisen through multiple reassortment events of MRV-3s from varied origins but are mostly closely related to bat, porcine and human MRV isolates.

MRVs are divided into three genotypes (MRV1-3), and genotyping is determined by segment S1. Our S1 tree (Figure 2) recapitulated the overall topology of other S1 strain phylogenies [18]. Based on the S1 phylogeny, the Human/SA395/SA/2017 MRV-3 strain belongs to an unassigned lineage of the MRV3 bearing the highest nucleotide identity, 93.64%, with the human Tahiti MRV3 strain (GenbankID: L37679) isolated from a child in Tahiti, French Polynesia, in 1960 [42] (Table 2). Given the close genetic relationship of SA397 and the Tahiti strain isolated in 1960, it is also likely that a fourth genotype of MRV-3 (3d) has been circulating in Southeast Asia and Oceania or southern Africa with limited genomic detection.

## 4. Discussion

In this study, we characterized two viruses, DENV1 and MRV, from a South African family that had recently returned from Thailand. Through metagenomics and phylogenetics, we identified the DENV1 strain as having originated from a common Southeast Asian lineage, DENV1I_K.1. The MRV strain was identified as a novel reassorted MRV-3c strain, with segments closely related to porcine, human and bat segments, suggesting that this and its S1 segment are most closely related to a human isolate from Tahiti, French Polynesia, from the 1960s. Investigating viruses detected in travelers returning from tropical areas, such as DENV and MRV, as described here, is crucial in understanding their origins, circulation, and potential for disease outbreaks. Such studies provide valuable insights into viral spread and evolution dynamics.

In 2019, Southeast Asia experienced a severe dengue outbreak, highlighting the persistent public health challenge posed by dengue fever in the region. Cambodia, for instance, reported over 10,000 cases annually, with more than 20,000 cases by July 2019 [48]. While DENV is not currently circulating in South Africa, several species of mosquitoes, including *Ae. aegypti*, are capable of vectoring DENV1 [49]. However, recent data on the vector competence of local mosquito populations for DENV1 are lacking. Only one study has indicated that South African colonies of *Ae. aegypti* could potentially promote DENV1/2 and yellow fever transmission, with the Durban population having the greatest vectorial capability [50].

In contrast to DENV1, the origin and circulation of MRVs are less well understood due to limited genetic epidemiology. Our analysis revealed high divergence in the L1 segment of the South African MRV strain compared to other MRV isolates in GenBank (88.57% similarity). The L1 segment was most closely related to a porcine MRV while other segments were closely related to bat and human segments. This suggests long-term circulation of MRV in Southeast Asia and Oceania, given the limited sampling and identity of MRV isolates from these regions.

Although the MRV was possibly acquired in Thailand, the lack of comprehensive genetic surveillance data prevents us from confirming this with certainty. The possibility of local acquisition of MRV in South Africa cannot be ruled out. It is unclear what the potential route of virus transmission was to the patient, as the focus of the epi information collected was on potential mosquito transmission, and all patients reported mosquito bites. No further information was collected at the time that could be used to elucidate potential routes of transmission. This uncertainty underscores the need for further research into the virus’s ecology, epidemiology, and evolution, emphasizing the necessity of comprehensive genetic surveillance to understand MRV’s geographic distribution.

This study also highlights the capacity of MRVs to circulate silently in human and animal populations. It contributes to the expanding database on MRV’s geographic distribution and molecular diversity.

Continued surveillance of returned travelers using metagenomic approaches is critical for the early identification of exotic viruses. Metagenomics allows for the comprehensive detection of known and novel pathogens, providing a powerful tool for monitoring the importation of viruses into non-endemic regions. Integrating metagenomic surveillance with traditional serological methods can enhance our ability to respond promptly to emerging infectious diseases, thereby mitigating their potential impact on public health.

## Figures and Tables

**Figure 1 viruses-16-01274-f001:**
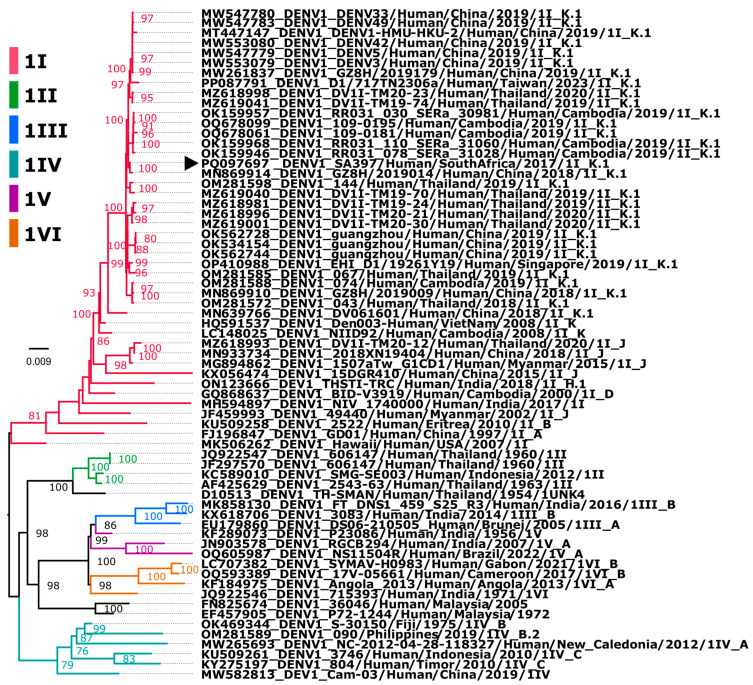
Phylogenetic analysis of imported DENV1 SA strain in 2017. The maximum likelihood (ML) tree was constructed based on aligned DENV1 strains with a nucleotide substitution model: TIM2+F+I+G4 with 1000 ultrafast bootstrap replicates. DENV1 strains are indicated as the 6 major genotypes (I–VI), highlighted in different colors with major and minor lineages if assigned on the label. Sequence accession number, country, and reported year are indicated, and the imported South African case is shown with arrowhead. Bootstrap support values are shown on nodes exceeding 75. The tree is rooted at the midpoint. Branch length corresponds to nucleotide substitutions per site.

**Figure 2 viruses-16-01274-f002:**
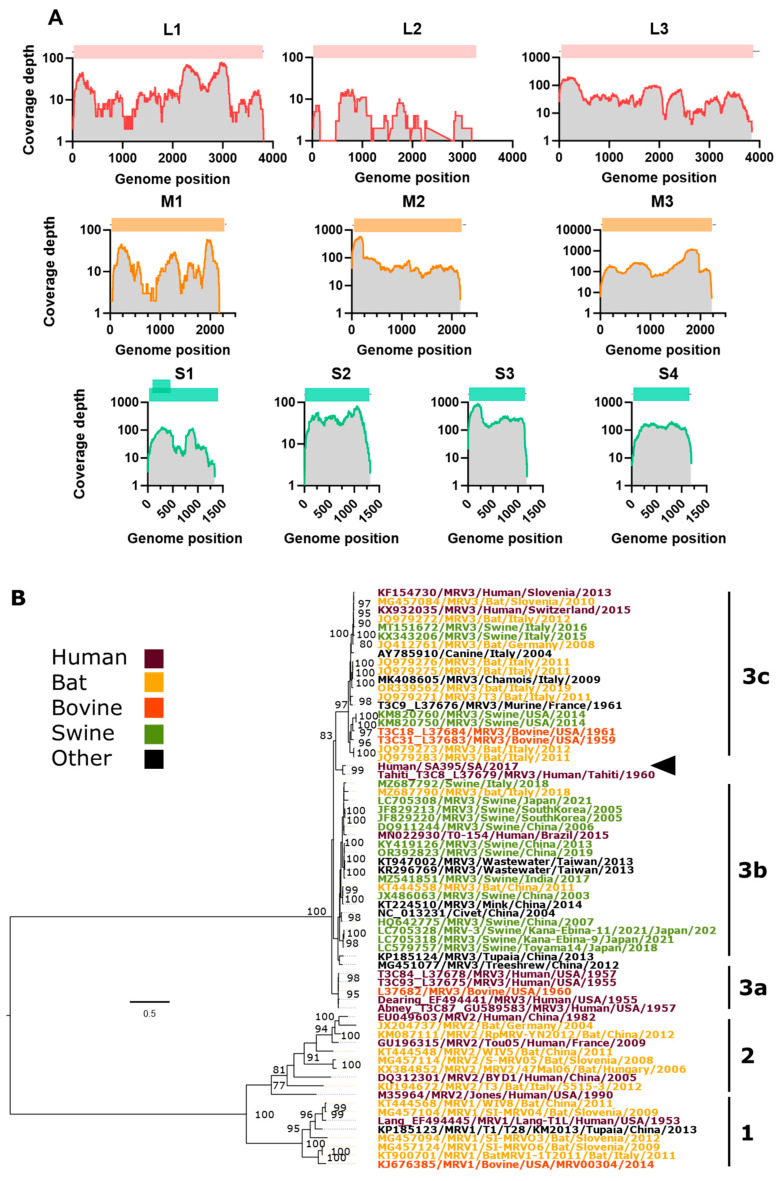
Genome organization and coverage, and phylogenetic inference of the S1 segment of the South African Human/SA395/SA/2017 MRV-3 strain. (**A**) Coverage and genome organization of the 10 MRV3 segments. (**B**) Consensus maximum likelihood phylogeny GTR+F+G4 model with a bootstrap of 1000 replicates. Accession number, host species, country and year are indicated for each strain. MRV-3 lineages are indicated in lowercase, a, b and c, and MRV-2 and MRV-1 serotypes are indicated. The South African Human/SA395/SA/2017 MRV-3 strain is indicated with an arrowhead. Bootstrap support values are shown on nodes exceeding 75. The tree is rooted at the midpoint. Branch length corresponds to nucleotide substitutions per site.

**Table 1 viruses-16-01274-t001:** Clinical features of patients recently returned from Thailand.

Date of Blood Collection	Case Number	Age	Sex	Symptoms	Clinical Features and Complications
17 July 2017	SA395	46	M	Onset (date not recorded): Fever (38.7 °C) without rash, headache, severe body aches	No blood hematology test results were reported. The patients were not hospitalized. No previous history of dengue or yellow fever vaccination. All patients reported mosquito bites.
	SA396	8	F	Onset (date not recorded): Fever (38.5 °C) without rash, severe body aches, nausea	
	SA397	45	F	Onset (16 July 2017): Fever (39 °C) without rash, severe body aches, nausea	

**Table 2 viruses-16-01274-t002:** Highest nucleotide identities for each gene segment of the imported Human/SA395/SA/2017 MRV-3 strain.

Segment Protein	Nucleotide (nt), Amino Acid (aa) Length	Closest Strain, GenBank, (%)	MRV Serotype	Location, Date	Host	Ref.
L1 λ3 protein	3822 nt 1267 aa	SHR-A JX415466 88.57%	MRV-1	China, 2011	Pig	[43]
L2 λ2 protein	2992/3918 nt (76.3%) 993/1299 aa (76.4%)	AP-151 MN022938 93.99%	MRV-3	Brazil, 2015	Human	[18]
L3 λ1 protein	3901 nt 1275 aa	RpMRV-YN2012 KM087107 92.72%	MRV-2	Yunnan province, China, 2012	*Rhinolophus pusillus* (bat)	[44]
M1 μ2 protein	2295 nt 736 aa	BatMRV/B19-02 MW582625 96.19%	MRV-1	Jeju Island, South Korea, 2019	*Miniopterus schreibersii* (bat)	[45]
M2 μ1 protein	2203 nt 708 aa	Abney T3C87 GU589581 93.64%	MRV-3	Washington, DC, USA, 1957	Human	[46]
M3 μNS protein	2241 nt 721 aa	BatMRV/B19-02 MW582627 96.90%	MRV-1	Jeju Island, South Korea, 2019	*Miniopterus schreibersii* (bat)	[45]
S1 σ1/σ1s protein	1380 nt σ1—455 aa σ1s—120 aa	Tahiti L37679 93.57%	MRV-3	Tahiti, French Polynesia, 1960	Human	[42]
S2 σ2 protein	1323 nt 427 aa	Lang L19774 98.11%	MRV-1	Ohio, USA, 1953	Human	[47]
S3 σNS protein	1198 nt 366 aa	BatMRV/B19-02 MW582630 96.79%	MRV-1	Jeju Island, South Korea, 2019	*Miniopterus schreibersii* (bat)	[45]
S4 σ3 protein	1196 nt 365 aa	SI-MRV07 MG999585 92.47%	MRV-3	Slovenia, 2017 *	Human	[17]

* Patient recently travelled from Thailand and Myanmar.

## Data Availability

Nucleotide sequences determined in this study were deposited in GenBank under the accession numbers MRV: PP953510-PP953519 and DENV1 PQ097697.

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
