# Peer review of "Detection of Dengue Virus 1 and Mammalian Orthoreovirus 3, with Novel Reassortments, in a South African Family Returning from Thailand, 2017"

_viruses, 2024, doi:10.3390/v16081274_

Round 1
Reviewer 1 Report
Comments and Suggestions for Authors
Petrus Jansen van Vuren et al reported a family of three members; a 46-year-old male, a 45-year-old female and their 8-year-old daughter, returned to South Africa from Thailand concurrent importation of two distinct viruses with different transmission routes in the same family with similar clinical presentations(a 45-year-old female infected dengue virus type 1 and . a 46-year-old male infected Mammalian orthoreovirus 3),which may be contributed to the understanding of the complexity of imported infection.
1. Mammalian reovirus is ubiquitous in mammals and can infect a wide variety of species worldwide. The incubation period of the virus in infected pigs was 72 hours. Please describe in detail the patient's symptoms and the time when the blood sample was collected in Materials and Methods. The author should exclude that the patient was infected after returned to South Africa from Thailand.
2. DENV can be classified into four distinct serotypes (DENV-1, DENV-2, DENV-3 and DENV-4) that share approximately 65% amino acid sequence similarity. DENV-1 can be divided into five genotypes (I, II, III, IV,and V) . Phylogenetic analysis of imported DENV1 SA strain in 2017 should added those five five genotypes reference strains.
3. Line 59, At the end of ‘travel’ , the words " in South Africa" should be added.
Author Response
Petrus Jansen van Vuren et al reported a family of three members; a 46-year-old male, a 45-year-old female and their 8-year-old daughter, returned to South Africa from Thailand concurrent importation of two distinct viruses with different transmission routes in the same family with similar clinical presentations(a 45-year-old female infected dengue virus type 1 and . a 46-year-old male infected Mammalian orthoreovirus 3),which may be contributed to the understanding of the complexity of imported infection.
- Mammalian reovirus is ubiquitous in mammals and can infect a wide variety of species worldwide. The incubation period of the virus in infected pigs was 72 hours. Please describe in detail the patient's symptoms and the time when the blood sample was collected in Materials and Methods. The author should exclude that the patient was infected after returned to South Africa from Thailand.
Response:
The details of patient symptoms are described in Table 1 in Results, in the Symptoms column. Description of column 1 of Table 1 has been changed to “Date of blood collection” for the sake of clarity. The following sentence was added in Results: Case Presentation “Date of onset of symptoms was only recorded for one patient (SA397) as 16 July (date of arrival back in South Africa), and blood samples were collected from all three patients on 17 July (one day after arrival in South Africa).”. The following statement is already included in the discussion to further address this point by Reviewer 1: “Although the MRV was likely acquired in Thailand, the lack of comprehensive ge-netic surveillance data prevents us from confirming this with certainty. The possibility of local acquisition of MRV in South Africa cannot be ruled out.”.
- DENV can be classified into four distinct serotypes (DENV-1, DENV-2, DENV-3 and DENV-4) that share approximately 65% amino acid sequence similarity. DENV-1 can be divided into five genotypes (I, II, III, IV,and V) .Phylogenetic analysis of imported DENV1 SA strain in 2017 should added those five five genotypes reference strains.
Response:
We appreciate that the phylogenetic inference presented in Figure 1 was minimal. We have now updated figure 1 to incorporate the updated DENV1 lineage status from the updated Hill et al., 2024 nomenclature. The tree now has 63 total DENV1 strains from with representatives from all genotypes. For the new phylogeny the topology of the and the position of the SA397 isolate within the major lineage 1I_K remains unchanged. We have updated the methods, figure legends and Data Availability Statement accordingly.
- Line 59, At the end of ‘travel’ , the words " in South Africa" should be added.
Response:
Corrected as suggested.
Reviewer 2 Report
Comments and Suggestions for Authors
Congratulations to the authors for their study of the subject, sagacity in investigating and carefully with co-infection of the sample, and for not leaving aside an importante finding.
It was an easy read with important results.
Below are my comments on the revision of the text.
There were a few typing, please find the corrections below. Nothing that detracted from the ease of understanding of the text. The manuscript was easy to read and very clear.
I would like to know why an with important information took 7 years to be submitted for publication?
Virological and entomological surveillance are the pillars for monitoring the epidemiological course of arboviruses and viruses. In today's world, there is no doubt that metagenomics is a valuable tool for identifying and classifying exotic and unknown viruses from different organisms present in a single sample. Once identified, they can disrupt pathways and future outbreaks and epidemics.
Therefore, I question the length of time it takes to submit such interesting and important data for publication.
Abstracta
Text corrections to lines:
Line 21 - headache, Line 30 - mammalian, Line 31 - analyses, Line 36 - co-circulation, Lines 39 and 94 - complexity, Line 40 - traveler.
Mixture of American and British English. Revise this part of the writing throughout the text.
Introduction
Explores the topic in a compact way, but with all the necessary information.
Some suggestions for better clarification for future readers follow below:
Line 56: The dengue virus is transmitted to humans primarily by the bite of female mosquitoes of the genus Aedes...
Line 57: Replace species with GENERO, species aegypti, albopictus, etc. Genus Aedes.
Materials and Methods
I have no objections to the methodology used to study the cases. I would use the same methods.
The methods used are all gold standard for virus isolation in human and vector samples, except for serology (hemagutination) in vectors.
Some typing correction.
Line 103 Hemaglutination
Line 105 Sinibis
Line 119 Manufaturar
Line 157 Fever not Fever
Resulta
The data presented enlightened the study, especially with the inclusion of the metagenomics technique, which could easily have been used with conventional techniques to detect arboviruses based on the classic symptoms and signs for arboviruses presented by the three patients and their reports of having been bitten by mosquitoes. The methods used in this approach were fundamental to understanding co-infection. There was a series of investigations using standard arbovirus diagnostic and virological techniques.
Phylogenetic analysis of the results and the percentages obtained showed clear phylogenetic evidence to distinguish the viruses found from DENV1, given the ubiquity and susceptibility of a wide range of hosts worldwide to these microorganisms. The composition of the data was robust and clear to clarify the case presented by the authors.
Discussion
Line 263 to bat and human segments suggests this.
Covering a general course of the manuscript and highlighting the results and clarifying the co-infection of the MRV virus, not excluding the possibility of acquiring MRV in South Africa. (My first question when I started reading the article was whether the patient SA397, from whom MRV was isolated, was already infected before arriving in Thailand) And in the discussion my doubt was very clear and clarified. (Lines 266 to 271)
I had to wait for the last lines.
The manuscrito ends by prioritizing and raising awareness of the inclusion of routine metagenomic techniques in public health surveillance programs, which can elucidate coinfections through genetic surveillance in regions of high transmission and multiple viral strains.
And metagenomics has fast and efficient screening for the investigation of samples from patients infected with unknown pathogens, not requiring the use of specific reagents for certain microorganisms, as is the case with conventional tests.
References
The authors have taken great care with the articles consulted and cited. All those cited in the text are included in the list of references. This is rarely the case. In general, references are missing, there are too many references, and some are not appropriate to the topic. None were missing, all were cited in accordance with the topic covered. It's good to take such care with the authors we quote in our articles, after all, they also helped us think and write the text by giving us their thoughts.
Thank you for providing such a pleasant read with such valuable information.
Comments on the Quality of English Language
The text has good English in some sentences there is a mix of British and American English, I made some corrections, but I suggest further revision.
Author Response
Congratulations to the authors for their study of the subject, sagacity in investigating and carefully with co-infection of the sample, and for not leaving aside an importante finding.
It was an easy read with important results.
Below are my comments on the revision of the text.
There were a few typing, please find the corrections below. Nothing that detracted from the ease of understanding of the text. The manuscript was easy to read and very clear.
I would like to know why an with important information took 7 years to be submitted for publication?
Virological and entomological surveillance are the pillars for monitoring the epidemiological course of arboviruses and viruses. In today's world, there is no doubt that metagenomics is a valuable tool for identifying and classifying exotic and unknown viruses from different organisms present in a single sample. Once identified, they can disrupt pathways and future outbreaks and epidemics.
Therefore, I question the length of time it takes to submit such interesting and important data for publication.
Response: We would like to thank the reviewer for the positive evaluation, and for asking this important question. We agree that the long delay in submitting the results is unfortunate. The reason for the long delay in submitting the manuscript for publication is due to a combination of reasons. One of the authors changed institutes shortly after the initial laboratory investigation; the COVID pandemic caused a significant delay as all authors switched focus of their research for a few years to COVD related research; one of the authors retired.
Abstracta
Text corrections to lines:
Line 21 - headache, Line 30 - mammalian, Line 31 - analyses, Line 36 - co-circulation, Lines 39 and 94 - complexity, Line 40 - traveler.
Mixture of American and British English. Revise this part of the writing throughout the text.
Response: all corrected
Introduction
Explores the topic in a compact way, but with all the necessary information.
Some suggestions for better clarification for future readers follow below:
Line 56: The dengue virus is transmitted to humans primarily by the bite of female mosquitoes of the genus Aedes...
Line 57: Replace species with GENERO, species aegypti, albopictus, etc. Genus Aedes.
Response: all corrected
Materials and Methods
I have no objections to the methodology used to study the cases. I would use the same methods.
The methods used are all gold standard for virus isolation in human and vector samples, except for serology (hemagutination) in vectors.
Some typing correction.
Line 103 Hemaglutination
Line 105 Sinibis
Line 119 Manufaturar
Line 157 Fever not Fever
Response: all corrected
Resulta
The data presented enlightened the study, especially with the inclusion of the metagenomics technique, which could easily have been used with conventional techniques to detect arboviruses based on the classic symptoms and signs for arboviruses presented by the three patients and their reports of having been bitten by mosquitoes. The methods used in this approach were fundamental to understanding co-infection. There was a series of investigations using standard arbovirus diagnostic and virological techniques.
Phylogenetic analysis of the results and the percentages obtained showed clear phylogenetic evidence to distinguish the viruses found from DENV1, given the ubiquity and susceptibility of a wide range of hosts worldwide to these microorganisms. The composition of the data was robust and clear to clarify the case presented by the authors.
Response: thank you
Discussion
Line 263 to bat and human segments suggests this.
Covering a general course of the manuscript and highlighting the results and clarifying the co-infection of the MRV virus, not excluding the possibility of acquiring MRV in South Africa. (My first question when I started reading the article was whether the patient SA397, from whom MRV was isolated, was already infected before arriving in Thailand) And in the discussion my doubt was very clear and clarified. (Lines 266 to 271)
I had to wait for the last lines.
The manuscrito ends by prioritizing and raising awareness of the inclusion of routine metagenomic techniques in public health surveillance programs, which can elucidate coinfections through genetic surveillance in regions of high transmission and multiple viral strains.
And metagenomics has fast and efficient screening for the investigation of samples from patients infected with unknown pathogens, not requiring the use of specific reagents for certain microorganisms, as is the case with conventional tests.
Response: corrected
References
The authors have taken great care with the articles consulted and cited. All those cited in the text are included in the list of references. This is rarely the case. In general, references are missing, there are too many references, and some are not appropriate to the topic. None were missing, all were cited in accordance with the topic covered. It's good to take such care with the authors we quote in our articles, after all, they also helped us think and write the text by giving us their thoughts.
Thank you for providing such a pleasant read with such valuable information.
Response: thank you
Reviewer 3 Report
Comments and Suggestions for Authors
See attached

Author Response
Detection of dengue virus 1 and mammalian orthoreovirus 3, with novel reassortments, in a South African family returning from Thailand, 2017 Van Vuren et al.
In this report, the authors describe the presence of dengue virus, mammalian orthoreovirus (MRV), and chikungunya virus in a South African family upon their return from Thailand. Characterization of the dengue virus (isolated from patient SA397) sequence revealed that it was of serotype 1. This patient acquitted the infection from a mosquito bite. Analysis of the serum sample from SA395 showed that the person was infected with MRV. The onset of symptoms was not given, making it difficult to conclude when the infection was acquired; the possible mode of infection was not discussed. It is also not clear why the chikungunya virus (from patient SA396) was not analyzed. Again, the onset of symptoms was not given.
Response: The date of onset of patient SA395 was not recorded on the case submission form submitted to the laboratory, and this is now stated more clearly in Table 1. The following sentences were added to the discussion to address the question about mode of infection: “It is unclear what the potential route of virus transmission was to the patient, as the focus of the epi information collected was on potential mosquito transmission, and all patients reported mosquito bites. No further information was collected at the time that could be used to elucidate potential routes of transmission.”. The chikungunya HAI results reported refer to serology, which is the routine screening done in the laboratory for antibodies to arboviruses. A clarification sentence has been added in the manuscript regarding the chikungunya HAI result.
Minor comment: Line 57, From 2000 to 2016, 176 laboratory-confirmed dengue cases …. , I believe this statement is only for South Africa. Please mention that.
Response: This has now been clarified in the text.
Round 2
Reviewer 1 Report
Comments and Suggestions for Authors
Petrus Jansen van Vuren et al reported a family of three members concurrent importation of two distinct viruses with different transmission routes in the same family with similar clinical presentations(a 45-year-old female infected dengue virus type 1 and . a 46-year-old male infected Mammalian orthoreovirus 3) returned to South Africa from Thailand ,which may be contributed to the understanding of the complexity of imported infection.The revised version have been sufficiently improved to publication.
Author Response
Petrus Jansen van Vuren et al reported a family of three members concurrent importation of two distinct viruses with different transmission routes in the same family with similar clinical presentations(a 45-year-old female infected dengue virus type 1 and . a 46-year-old male infected Mammalian orthoreovirus 3) returned to South Africa from Thailand ,which may be contributed to the understanding of the complexity of imported infection.The revised version have been sufficiently improved to publication.
Response: Thank you.
Reviewer 3 Report
Comments and Suggestions for Authors
Detection of Dengue virus 1 and Mammalian orthoreovirus 3, 2 with novel reassortments, in a South African family returning 3 from Thailand, 2017
van Vuren et al.
The revised version has been improved. However, it is still not clear (as suggested in the Discussion) whether MRV was imported. Therefore, I suggest not emphasizing the importation of MRV. The manuscript still needs proofreading.
Line 272 and 274, Ae. aegypti should be italicized.
Line 281, a period (full stop) is missing.
Comments on the Quality of English LanguageThe manuscript needs proofreading.
Author Response
Detection of Dengue virus 1 and Mammalian orthoreovirus 3, 2 with novel reassortments, in a South African family returning 3 from Thailand, 2017
van Vuren et al.
The revised version has been improved. However, it is still not clear (as suggested in the Discussion) whether MRV was imported. Therefore, I suggest not emphasizing the importation of MRV. The manuscript still needs proofreading.
Response: The manuscript has been edited to reduce the emphasis on importation. The manuscript has been proofread.
Line 272 and 274, Ae. aegypti should be italicized.
Response: corrected
Line 281, a period (full stop) is missing.
Response: corrected